# Luminescence properties of Ge and Ge-Si structures on silicon-based microcavities

**Xiaoyu Zhou[1,2], Ziyang Zhou[1], Wenqian Wu[1], Weiye Yang[1,3], Hongyan Peng[1,3], Shihua Zhao** [1,3]*

**1** College of Physics and Electronic Engineering, Hainan Normal University, Haikou, China, **2** The Innovation Platform for Academicians of Hainan Province, Haikou, China, **3** Jilin Sino-Microelectronics Co. Ltd, Jilin, Jilin, China

* zsh@hainnu.edu.cn

## Abstract

Utilizing laser etching technology under specific laser parameters (laser energy 250 mJ, repetition rate 1 Hz, pulse width 30 ns, spot diameter 150 μm), single-crystal silicon was etched, followed by the deposition of materials onto silicon-based microcavity samples using the pulsed laser deposition method. The Ge-coated microcavity samples were characterized using SEM and EDS, revealing cluster formations at the edges of the microcavities with the presence of germanium within these clusters. Photoluminescence spectroscopy identified characteristic luminescence peaks due to transverse optical (TO) phonon vibrations at 695 nm in the Ge-coated microcavity samples. Subsequently, the Ge-coated microcavity samples were subjected to high-temperature annealing at 1000 °C. After annealing for 30 min, a sharp luminescence peak at 700 nm associated with Ge-O bonds was observed. Finally, a layer of Si was deposited over the Ge-coated samples using pulsed laser deposition, resulting in a Ge-Si bilayer structure. After annealing at 1000 °C, this bilayer structure exhibited two distinctive peaks at 700 nm and 940 nm, with the former being a sharp peak due to Ge-O bonds and the latter due to the formation of Si-Si bonds within the clusters.

## Introduction

Silicon has rapidly advanced as a material in microelectronics. However, due to its indirect bandgap and relatively large bandgap width ($E_g$ = 1.12 eV), its luminous efficiency is low. The efficiency of energy level transitions in silicon is significantly lower than that in direct bandgap semiconductor materials, making it challenging to achieve high-efficiency silicon-based light-emitting devices. Consequently, silicon's limited application in optoelectronics has made the study of its luminescent properties a hot topic in the fields of semiconductors and optoelectronics [1–2].

In 1990, Canham made a groundbreaking discovery: porous silicon can exhibit photoluminescence (PL) at room temperature (21 °C) [3–4]. This pioneering finding

**Data availability statement:** All relevant data are within the paper and its Supporting Information files.

**Funding:** This work was supported by the National Natural Science Foundation of China (U1704145), the Hainan Provincial Natural Science Foundation of China (522MS062), and the specific research fund of the Innovation Platform for Academicians of Hainan Province (YSPTZX202207).

**Competing interests:** The authors have declared that no competing interests exist.

marked a significant breakthrough in the field of silicon-based luminescent materials, sparking a surge in research interest [5]. Owing to its simple preparation process, efficient light absorption, and luminescence properties, porous silicon garnered considerable attention, becoming a popular research subject [6]. In 2004, Fabio and colleagues fabricated Si nanoclusters within $SiO_2$ by thermally annealing $SiO_x$ thin films prepared via plasma-enhanced chemical vapor deposition. They systematically compared the optical and structural characteristics of these nanoclusters, demonstrating that the photoluminescence properties depended on the amorphous and crystalline clusters [7]. In 2006, Negro and others employed similar deposition and annealing techniques to successfully fabricate silicon nitride thin films, measuring their external quantum efficiency and studying the impact of temperature on their luminescence [8]. In 2014, Huang and colleagues explored the photoluminescence and electroluminescence kinetics of Yb-coated nano-silicon. They observed stimulated emission peaks near 700 nm on Yb-coated silicon quantum dots and measured enhanced EL peaks in the wavelength range of 1300 nm to 1650 nm on Yb-coated Si thin films. They designed a Si-Yb quantum cascade laser, where Si-Yb bonding on the raised surfaces of Si films allowed manipulation of the emission wavelength into the optical communication window, offering new insights into silicon-based luminescence research [9]. In 2016, Dejene and others utilized pulsed laser deposition (PLD) to coat silicon substrates with cerium, studying the surface structure and luminescence performance. Their experiments revealed that the increase in PL intensity was associated with a rougher surface [10].

In 2018, Yun et al. utilized an alternating single and dual target sputtering technique to deposit nanoscale silicon multilayer films at low temperatures. PL spectroscopy revealed that by controlling the thickness of the nanolayers within the multilayer structure, intense luminescence was achieved. This phenomenon was attributed to complex quantum confinement effects and defect state emission mechanisms [11]. In the same year, Chang Gengrong and his team produced amorphous silicon-rich silicon carbide germanium thin films using magnetron sputtering. Subsequent high-temperature annealing resulted in the formation of anisotropic silicon nanocrystals. Studies indicated that the anisotropic strain energy induced the formation of multifaceted silicon nanocrystals, significantly altering their energy level structure. PL emission spectra were observed at 2.57 eV and 2.64 eV, with a notable increase in the light absorption band [12]. In 2020, Iwaya et al. prepared supersaturated silicon samples in silicon dioxide using PLD in an oxygen atmosphere. Post-annealing at high temperatures led to the formation of luminescent Si nanocrystals embedded within the $SiO_2$. Effective photoluminescence was observed in these annealed samples [13]. That same year, Anastasiya and colleagues conducted pulsed laser ablation experiments on silicon targets in a helium (He) - nitrogen (N) gas mixture, fabricating silicon (Si) nanocrystal thin films with strong photoluminescence properties. Different luminescent characteristics were observed by varying the He/N concentrations [14]. In 2022, Jian Yan and her team employed an electrochemical anodization method to produce porous silicon on a small scale. The results showed that the luminescence intensity of the porous silicon first increased and then

decreased with the lengthening of the etching time, inversely correlating with the current density. The fluorescence peak shifted towards red with an increase in hydrofluoric acid concentration in the electrolyte solution [15]. In 2023, Zhang et al. explored the luminescent and nonlinear optical properties of silicon quantum dot multilayer films with various quantum dot diameters using continuous-wave and picosecond lasers. The results indicated that both the linear and nonlinear optical performances of the multilayer films were closely related to the size of the quantum dots, with the photoluminescence intensity increasing by approximately 32 times [16]. In 2024, Wang and others grew Ge layers on Si (100) substrates, achieving optical responsivities of 0.51 and 0.17 A/W at 1310 nm and 1550 nm, respectively, and also demonstrated their superior electrical properties [17]. In recent years, the exceptional performance of small-sized silicon materials has made them a focal point of research, exploring their luminescence mechanisms, performance enhancements, and potential applications, aiming to provide valuable references for subsequent studies. To enhance the luminescent performance of silicon-based materials, researchers have explored various methods for preparing silicon-based luminescent structures, such as chemical vapor deposition [18], vacuum evaporation [19], magnetron sputtering [20], and PLD [21–22].

Chemical vapor deposition is one of the most widely applied methods for fabricating silicon-based thin films, known for its low operational costs, good controllability, and high uniformity of the films. However, this method can lead to the emission of harmful gases during the fabrication process, requires sophisticated equipment, and is prone to impurity inclusion. Magnetron sputtering is a straightforward method that can produce high-purity silicon-based thin films but often results in films of relatively poor quality, lacks precise control over the film properties, and involves high equipment costs. Vacuum evaporation, which uses sublimation or evaporation of silicon-based materials to create films, offers the advantages of rapid production and good film density. Nonetheless, it demands high-end equipment, is costly, and tends to generate pollution during production. Consequently, this paper employs PLD, a method known for its strong controllability and minimal pollution, to prepare silicon-based microcavity luminescent samples. Using nanosecond pulsed laser irradiation of monocrystalline silicon, the samples are prepared and then layered with Ge and Ge-Si using PLD. The luminescent properties of these samples, both pre and post annealing under various conditions, are analyzed using a Raman spectrometer, providing valuable insights for further research and development of silicon-based luminescent materials.

## Experiments

### Experimental equipment

The experiments utilized P-type (100) oriented monocrystalline silicon with a resistivity of 10–20 Ω·cm, a self-built nanosecond pulsed laser system based on electro-optic Q-switching, a Shenyang KeYou L-MBE450E laser molecular beam epitaxy system, a Hitachi S-4800 scanning electron microscope (SEM) from Japan, a tube furnace made by Shanghai Lichen Bangxi Instrument Technology Co., Ltd., The fabricated films were characterized using various techniques. First, X-ray diffrac-tion (XRD) (Rigaku Ultima-IV) with Cu Ka radiation (wavelength λ = 1.5406 A) was used todetermine the crystalline structure of the films over a 20 range of 20° to 80°. Second, a JEOLJSM-7100F scanning electron microscope was used to observe the surface morphology. In this study, a Renishaw Raman spectrometer was used to test the PL spectroscopy of the sample, which had an excitation light of 532 nm, a spectral detection range of 547–900 nm, and an excitation power of up to 50 mW

### Preparation methods and procedures

P-type monocrystalline silicon samples with (100) orientation and a resistivity of 10–20 Ω·cm were prepared. In this experiment, silicon wafers were immersed in a mixed solution of 30 milliliters of 95% concentrated ethanol and 30 milliliters of 98% concentrated acetone for 5 minutes to remove surface organics. Afterward, the silicon wafers were cleaned with deionized water using an ultrasonic cleaner for 15 minutes. The processed samples were then etched using a pulsed laser etching technique. The laser parameters employed were an energy of 250 mJ, a pulse width of 30 ns, a repetition

rate of 1 Hz, a spot diameter of 150 µm, and an irradiation time of 4 s, keep room temperature at 21 °C. This process created microcavity structures with a diameter of approximately 150 µm, as shown in S1 Fig (a) in S1 File. The photoluminescence spectroscopy of the sample is displayed in S1 Fig(b) in S1 File, featuring a characteristic luminescence peak at 700 nm attributed to the silicon-based microcavity. The sharp peak at 547 nanometers is attributed to the TO phonon in crystalline silicon (c-Si), the peak at 540 nanometers corresponds to 2TA in c-Si, and the peak at 563 nanometers represents 2TO in c-Si. There is no correlation peak with the silicon dioxide at 560 nm [23].

After etching, the silicon-based microcavity samples were further processed through PLD. A COMPex201 excimer laser operating at a wavelength of 248 nm, with a repetition rate of 2 Hz and an energy of 230 mJ, was used to directly irradiate the Ge target material via an optical path and laser window inside the molecular beam epitaxy chamber. The target material absorbed the laser energy, rapidly increasing the temperature at the spot of irradiation, causing the Ge to evaporate. The evaporated material formed a plasma, which, under continued interaction with the laser beam, developed into an elongated plasma plume. This facilitated the deposition of Ge particles onto the substrate, resulting in a single layer of Ge-coated silicon-based microcavity samples. Following the same procedure, the laser was aimed at a Si target to deposit a Si layer on the microcavity samples. A further layer of Ge was then deposited over the Si-coated samples, producing a Ge-Si bilayer structure. The entire process was conducted in a vacuum environment at $5 \times 10^{-4}$ Pa to ensure that the plasma-assisted deposition was free from external gaseous interference. The deposited samples were then placed in a tube furnace and annealed at 101.325 kPa atmospheric pressure at 1000 °C for different periods of time in an atmospheric atmosphere to study the luminescence properties of the samples with different deposited elements and annealing times.

## Results and discussion

### Study of luminescence performance in non-annealed microcavity germanium-coated samples

Based on the luminescence of silicon-based microcavities, germanium-coated silicon-based microcavity samples were prepared using the PLD method. S2 Fig(a) in S1 File shows the SEM image at the edge of the microcavity, and S2 Fig(b) in S1 File is an enlarged view of the boxed area in S2 Fig(a) in S1 File, where densely clustered particles can be observed at the edge of the microcavity. S2 Fig(c) in S1 File presents the EDS spectrum of the same boxed area from S2 Fig(a) in S1 File, showing the distribution of Si and Ge elements on the surface, with a larger proportion of silicon due to the silicon substrate.

X-ray diffraction (XRD) was performed on the samples. S3 Fig(a) in S1 File shows the XRD pattern, revealing diffraction peaks at 27°, 31°, 45°, and 69°. The XRD pattern reveals the coexistence of Ge (110) and Ge (220) crystal planes, as well as Si (200) and Si (400) crystal planes on the surface of the silicon-based microcavity. In the XRD pattern, there are also two low peaks located at 57° and 75°, indicating the generation of weak germanium dioxide crystal phase during laser etching. S3 Fig(b) in S1 File displays the photoluminescence spectroscopy at the edge of the microcavity clusters post-deposition, where a broad spectrum stimulated emission phenomenon around 695 nm is observed [24], similar to the luminescent effect of silicon microcavities. This broadband emission is attributed to the characteristic luminescence produced by transverse optical (TO) phonon vibrations at the silicon-based clusters at the edge of the microcavity [25].

### Study of luminescence performance in annealed microcavity germanium-coated samples

The germanium-coated silicon-based microcavity samples were subjected to high-temperature annealing at 1000 °C for varying durations, resulting in annealed samples. XRD analyses were conducted on these samples, with S4 Fig(a-f) in S1 File showing the XRD patterns for annealing times of 10 min, 20 min, 30 min, 40 min, 50 min, and 60 min, respectively. The patterns reveal diffraction peaks at 27°, 31°, 45°, 57°, 62°, and 69°. Peaks at 27° and 45° correspond to Ge (110) and Ge (220) crystal faces, respectively, with sharp and narrow peaks indicating high crystallinity. Peaks at 57° and 62° correspond to $GeO_2$ (210) and $GeO_2$ (113) crystal faces. Peaks at 31° and 69° correspond to the Si (200) and Si (400) crystal faces, indicating the presence of Ge and Si elements in the microcavity samples deposited on the silicon substrate, with

possible incorporation of Ge elements. As annealing time increases, the peak intensity ratios exhibit a trend of increasing and then decreasing, reaching a maximum around 40–50 min. Therefore, while maintaining a consistent annealing temperature, increasing the annealing duration enhances the sample quality up to a point, beyond which the peak intensity ratio decreases and the sample quality deteriorates.

The photoluminescence spectroscopy of the annealed samples are depicted in S5 Fig in S1 File, where panel (a) shows the spectrum of the sample annealed for 10 min, featuring a broad peak around 700 nm. This peak exhibits a slight redshift compared to the spectrum of the unannealed sample, an effect attributed to the annealing process. Panel (b) presents the photoluminescence spectroscopy of the sample annealed for 20 min, which still displays the characteristic broad peak around 700 nm. In panel (c), the spectrum for the 30-minute annealed sample also shows a broad peak at approximately 700 nm, but with an additional sharp peak at this wavelength. This sharp peak results from increased crystallization due to longer annealing times, where the high-temperature treatment facilitates a full reaction between germanium (Ge) in the clusters and oxygen, forming Ge-O bonds. Thus, the sharp peak at 700 nm is due to localized luminescence from Ge-O bonds within these clusters. Huang W Q had employed a quantum dot confinement model to calculate the energy gap structure of germanium oxide nanocrystal clusters, and utilized the Monte Carlo method to simulate the PL spectrum and the corresponding size distribution of the germanium oxide nanocrystal clusters. The calculated results are in agreement with our experimental findings [26]. Panel (d) illustrates the photoluminescence spectroscopy of the sample annealed for 40 min, where the sharp peak at 700 nm persists, indicating the presence of localized luminescence peaks from Ge-O bonds, with an increase in luminescence intensity as the annealing time extends. Panel (e) shows the spectrum for the 50-minute annealed sample, where the localized luminescence peak at 700 nm remains but with a luminescence intensity lower than that of the 40-minute sample. In panel (f), the spectrum of the 60-minute annealed sample also displays this localized luminescence peak at 700 nm, but with an intensity less than that of the 50-minute sample. As the annealing time changes, it can be observed through XRD and Raman spectroscopy that the TO peak of Ge-O gradually increases, and the crystallization performance improves until a certain annealing time is reached. The Ge-O cluster structure is destroyed, and the crystallization performance deteriorates. In the annealed samples, XRD peaks of germanium oxide nanocrystals and silicon (400) crystals dominate, and the percentage of silicon atoms decreases due to the dominant position of oxygen atoms in the composition. As the annealing time increases, the degree of crystallization of the samples also increases, enhancing the sample quality and luminescence effect, peaking at 40 min. However, excessively long annealing times lead to a reduction in peak intensity ratio, deteriorating cluster quality and luminescence effects. The sample surface is populated with many small-sized germanium crystal clusters. When the sample is processed under annealing conditions, the germanium crystals on the surface react with oxygen atoms. After the oxygen atoms diffuse into the germanium layer, they coalesce into Ge-O clusters above the oxide layer. However, annealing for a duration longer than a certain threshold can disrupt the cluster structure of Ge-O. These findings on the luminescence mechanisms and performance of silicon-based structures provide valuable insights for future research on silicon-based light-emitting devices.

Given the optimal luminescence observed at 40 min of annealing, morphological characterization and energy-dispersive X-ray spectroscopy (EDS) tests were conducted on the sample annealed for 40 min. S6 Fig(a) in S1 File shows the SEM image of the sample, revealing cluster crystallization at the microcavity edges and an increase in size due to annealing. S6 Fig(b) in S1 File presents the EDS spectrum of the same sample, indicating the presence of Si, Ge, and O at the microcavity edges, with an increased oxygen content compared to the unannealed sample. This supports the likelihood of Ge-O bond formation within the edge clusters.

## Study on the luminescence performance of non-annealed germanium-silicon coated microcavity samples

Silicon-based microcavity samples coated with germanium were prepared using pulsed laser deposition (PLD). Subsequently, Si targets were deposited onto these germanium-coated silicon-based microcavity samples using PLD again,

resulting in an additional layer of Si structure on the germanium-coated microcavity samples. This process yielded microcavity samples with a Ge-Si bilayer structure. S7 Fig(a) in S1 File presents the SEM images of the samples, with S7 Fig(b) in S1 File showing an enlarged view of the boxed area in S7 Fig(a) in S1 File, where the cluster structures at the edge of the microcavity can be observed. S7 Fig(c) in S1 File shows the EDS spectrum of the boxed area in S7 Fig(a), with the EDS analysis detecting the presence of silicon and germanium elements on the surface. S8 Fig(a) in S1 File displays the XRD patterns of the bilayer samples, showing diffraction peaks at 31°, 45°, and 69°. The peak at 45° corresponds to the Ge (220) plane, while the peaks at 31° and 69° correspond to the Si (200) and Si (400) planes, respectively, with the peak at 69° being particularly sharp and narrow, indicating a high degree of crystallinity. S8 Fig(b) in S1 File shows the photoluminescence spectroscopy of the bilayer samples, with a broad stimulated luminescence peak appearing at 900 nm, attributed to the formation of Si-Si bonds within the clusters due to the incorporation of silicon.

### Study on the Luminescence Performance of Annealed Germanium-Silicon Coated Microcavity Samples

The Ge-Si bilayer structure samples were subjected to high-temperature annealing at 1000 °C for varying durations using a tube furnace. S9 Fig in S1 File presents the XRD patterns of the bilayer structure samples after different annealing times. The XRD pattern for a 10-minute anneal, shown in S9 Fig(a) in S1 File, displays diffraction peaks at 27°, 31°, 45°, and 57°, with the peaks at 27° and 45° corresponding to the Ge (100) and Ge (220) planes, respectively, the peak at 31° corresponding to the Si (200) plane, and the peak at 57° corresponding to the $GeO_2$ (210) plane. Figures 9(b-e) show the XRD patterns for annealing durations of 20, 30, 40, and 50 min, respectively. These images reveal that with increasing annealing time, the $GeO_2$ (203) plane appears at 66°, and the peak intensities and shapes improve, with a decrease in full width at half maximum. These spectral changes occur because when the samples are not annealed or annealed for shorter durations, the atoms deposited on the substrate surface have lower energy, leading to smaller grain sizes. As the annealing time increases, the atoms gain sufficient energy to migrate to optimal locations on the sample, resulting in larger grain sizes and the formation of better-quality clusters. S9 Fig(f) in S1 File notes that annealing for 60 min results in a decrease in peak intensity relative to the background, leading to degraded sample quality and diminished luminescence effects.

S10 Fig in S1 File presents the photoluminescence spectroscopy of bilayer structure samples post-annealing. S10 Fig(a) in S1 File shows the photoluminescence spectroscopy after 10 min of annealing, with peaks at 700 nm and 930 nm. The peak at 700 nm corresponds to localized luminescence from Ge-O bonds, while the peak at 930 nm results from luminescence due to the formation of Si-Si bonds within clusters, exhibiting a redshift compared to the unannealed sample. Figures 10(b-f) depict photoluminescence spectroscopy after annealing for 20, 30, 40, 50, and 60 min, respectively. Peaks were observed at 700 nm and 940 nm in these spectra. The 700 nm peak corresponds to localized luminescence from Ge-O bonds, while the 940 nm peak results from luminescence due to the formation of Si-Si bonds within clusters, with a redshift observed compared to the 10-minute annealed sample. The deposition process is carried out under vacuum conditions, while the annealing process is conducted in an atmospheric environment. The source of oxygen during annealing is the oxygen present in the atmosphere, which reacts with the surface Ge under high temperature to form a cluster structure of Ge-O. The discovery and optimization of luminescence mechanisms and performance in various structured silicon-based microcavities provide valuable references for further research on silicon-based luminescent devices.

The optimal luminescence was observed at 40 min of annealing; thus, the Ge-Si bilayer structure sample annealed for 40 min underwent SEM morphological characterization and EDS spectral analysis. S11 Fig(a) in S1 File displays the SEM image of the sample annealed for 40 min, showing cluster crystallization at the microcavity edges and an increase in size due to annealing. S11 Fig(b) in S1 File shows the EDS spectrum of the region framed in S10 Fig(a) in S1 File, revealing the presence of Si, Ge, and O elements at the microcavity edges with an increased oxygen content compared to the unannealed sample, suggesting the possible formation of Ge-O bonds within the clusters.

## Conclusion

This paper primarily employs laser etching and PLD to prepare Ge and Ge-Si structured samples on silicon-based microcavities, investigating their luminescence properties through photoluminescence spectroscopy. Research findings indicate that post-Ge coating, the silicon-based microcavity samples exhibit a broad stimulated luminescence near 695 nm, attributed to the TO phonon vibrations from cluster structures at the microcavity edges. After annealing for 30 min, a sharp characteristic luminescence peak appears at 700 nm due to the localized luminescence from Ge-O bonds within clusters. The Ge-Si bilayer samples display a broad luminescence peak at 900 nm, generated by the incorporation of silicon forming Si-Si bonds within the clusters. Post-annealing, the spectra reveal characteristic luminescence peaks at both 700 nm and 940 nm. The peak at 700 nm corresponds to the localized luminescence from Ge-O bonds, whereas the peak at 940 nm results from the luminescence due to the formation of Si-Si bonds, with a notable redshift phenomenon compared to the non-annealed samples. These luminescence properties enable the silicon-based microcavities to exhibit unique stimulated emission wavelengths. The study not only paves new paths for cutting-edge applications such as lasers, light-emitting diodes, and solar cells but also provides invaluable references for further development of silicon-based luminescent materials and devices.

## Supporting information

**S1 File.** S1 Fig. Characterization of the microcavity samples: (a) a microscopic image of the microcavity; (b) the photoluminescence spectroscopy of the sample. S2 Fig. Characterization of the microcavity germanium-coated samples: (a) Secondary electron scanning electron microscope image at the edge of the microcavity, (b) enlarged view of part (a), (c) EDS spectrum at the edge of the microcavity. S3 Fig. Characterization of the germanium-coated microcavity samples: (a) XRD pattern of the sample, (b) photoluminescence spectroscopy of the sample. S4 Fig. XRD patterns of the germanium-coated microcavity samples at different annealing times: (a) after annealing for 10 min, (b) after annealing for 20 min, (c) after annealing for 30 min, (d) after annealing for 40 min, (e) after annealing for 50 min, (f) after annealing for 60 min. S5 Fig. Photoluminescence spectroscopy of germanium-coated microcavity samples at different annealing times: (a) after annealing for 10 min, (b) after annealing for 20 min, (c) after annealing for 30 min, (d) after annealing for 40 min, (e) after annealing for 50 min, and (f) after annealing for 60 min. S6 Fig. Characterization of the germanium-coated microcavity sample annealed for 40 min: (a) SEM image and (b) EDS spectrum. S7 Fig. Characterization of the Ge-Si coated microcavity samples: (a) the SEM image at the edge of the microcavity, (b) an enlarged view of (a), and (c) the EDS spectrum at the edge of the microcavity. S8 Fig. Characterization of the Ge-Si coated microcavity samples: (a) the XRD patterns, and (b) the photoluminescence spectroscopy of the samples. S9 Fig.XRD patterns of Ge-Si coated microcavity samples after different annealing times: (a) XRD patterns after 10 minutes of annealing, (b) XRD patterns after 20 minutes of annealing, (c) XRD patterns after 30 minutes of annealing, (d) XRD patterns after 40 minutes of annealing, (e) XRD patterns after 50 minutes of annealing, (f) XRD patterns after 60 minutes of annealing. S10 Fig. Photoluminescence spectroscopy of Ge-Si Microcavity Samples at Different Annealing Times: (a) the spectrum after 10 min; (b) after 20 min; (c) after 30 min; (d) after 40 min; (e) after 50 min; (f) after 60 min. S11 Fig. Characterization of the Ge-Si microcavity sample annealed for 40 min: (a) SEM image of the sample; (b) EDS image from the annealed sample.
(DOCX)

## Author contributions

**Conceptualization:** Ziyang Zhou, Wenqian Wu, Weiye Yang, Hongyan Peng.

**Writing – original draft:** Xiaoyu Zhou.

**Writing – review & editing:** Xiaoyu Zhou, Shihua Zhao.

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
