## [Decision Letter · Decision Letter 0]

6 Dec 2024

PONE-D-24-46249Luminescence Properties of Ge and Ge-Si Structures on Silicon-Based MicrocavitiesPLOS ONE

Dear Dr. Zhao,

Thank you for submitting your manuscript to PLOS ONE. After careful consideration, we feel that it has merit but does not fully meet PLOS ONE’s publication criteria as it currently stands. Therefore, we invite you to submit a revised version of the manuscript that addresses the points raised during the review process.

We look forward to receiving your revised manuscript.

Kind regards,

Mallikarjuna Reddy Kesama, Ph.D.

Academic Editor

PLOS ONE

Journal Requirements:

2. Thank you for stating the following financial disclosure: [This work was supported by the National Natural Science Foundation of China (U1704145)�the Hainan Provincial Natural Science Foundation of China (522MS062), and the specific research fund of the Innovation Platform for Academicians of Hainan Province (YSPTZX202207).]. Please state what role the funders took in the study. If the funders had no role, please state: "The funders had no role in study design, data collection and analysis, decision to publish, or preparation of the manuscript." If this statement is not correct you must amend it as needed. Please include this amended Role of Funder statement in your cover letter; we will change the online submission form on your behalf.

3. Thank you for stating the following in the Acknowledgments Section of your manuscript: [c This work was supported by the National Natural Science Foundation of China (U1704145), the Hainan Provincial Natural Science Foundation of China (522MS062), and the specific research fund of the Innovation Platform for Academicians of Hainan Province (YSPTZX202207).] We note that you have provided funding information that is not currently declared in your Funding Statement. However, funding information should not appear in the Acknowledgments section or other areas of your manuscript. We will only publish funding information present in the Funding Statement section of the online submission form. Please remove any funding-related text from the manuscript and let us know how you would like to update your Funding Statement. Currently, your Funding Statement reads as follows: [This work was supported by the National Natural Science Foundation of China (U1704145)�the Hainan Provincial Natural Science Foundation of China (522MS062), and the specific research fund of the Innovation Platform for Academicians of Hainan Province (YSPTZX202207).] Please include your amended statements within your cover letter; we will change the online submission form on your behalf.

4. We note that your Data Availability Statement is currently as follows: [All relevant data are within the manuscript and its Supporting Information files.] Please confirm at this time whether or not your submission contains all raw data required to replicate the results of your study. Authors must share the “minimal data set” for their submission. PLOS defines the minimal data set to consist of the data required to replicate all study findings reported in the article, as well as related metadata and methods (https://journals.plos.org/plosone/s/data-availability#loc-minimal-data-set-definition). For example, authors should submit the following data: - The values behind the means, standard deviations and other measures reported; - The values used to build graphs; - The points extracted from images for analysis. Authors do not need to submit their entire data set if only a portion of the data was used in the reported study. If your submission does not contain these data, please either upload them as Supporting Information files or deposit them to a stable, public repository and provide us with the relevant URLs, DOIs, or accession numbers. For a list of recommended repositories, please see https://journals.plos.org/plosone/s/recommended-repositories. If there are ethical or legal restrictions on sharing a de-identified data set, please explain them in detail (e.g., data contain potentially sensitive information, data are owned by a third-party organization, etc.) and who has imposed them (e.g., an ethics committee). Please also provide contact information for a data access committee, ethics committee, or other institutional body to which data requests may be sent. If data are owned by a third party, please indicate how others may request data access.

Additional Editor Comments:

This manuscript presents a detailed investigation into the luminescence properties of Ge and Ge-Si structures on silicon-based microcavities, employing innovative methods like laser etching and pulsed laser deposition (PLD). The study is comprehensive, with robust experimental design and clear data presentation, highlighting key findings such as dual luminescence peaks at 700 nm and 940 nm. However, the manuscript would benefit from a more concise introduction that sharpens its focus and highlights the novelty of the research. The results and discussion sections could be strengthened by integrating comparisons with existing literature and providing deeper insights into the mechanisms driving the observed luminescence. Additionally, improvements in figure clarity, language precision, and reference formatting are necessary to enhance readability and impact. Overall, the research has significant implications for advancing silicon-based luminescent materials, and addressing these issues will further align the manuscript with the expectations of a high-impact audience.

Reviewers' comments:

Reviewer's Responses to Questions

**Comments to the Author**

1. Is the manuscript technically sound, and do the data support the conclusions?

Reviewer #1: Partly

Reviewer #2: Partly

2. Has the statistical analysis been performed appropriately and rigorously?

Reviewer #1: N/A

Reviewer #2: N/A

3. Have the authors made all data underlying the findings in their manuscript fully available?

Reviewer #1: Yes

Reviewer #2: Yes

4. Is the manuscript presented in an intelligible fashion and written in standard English?

Reviewer #1: Yes

Reviewer #2: Yes

5. Review Comments to the Author

Reviewer #1: The manuscript titled “Luminescence Properties of Ge and Ge-Si Structures on Silicon-Based Microcavities” is systematic and well written. The laser etching, subsequentially PLD process Ge and Si-Ge growth is novel technique. However, the argument of Luminescence significance in the report is not confirmed. It is rather micro-Raman analysis of SiGe nanocrystal. Authors tried to establish the strained SiGe nanolayers optical significance on Ge coated microcavities. Without annealing, the grains orientation shown by SEM and XRD analysis are realized polycrystalline properties. Generally, high temperature there is a possibility of Si diffusion and deteriorates crystalline properties. Moreover, high energy LASER application and greater oxygen contamination can make it polycrystalline. The micro–Raman TO broader peak seems to display it. Is there any evidence to retain its crystallinity? As grown SiGe nanostructure clustering is less dominant while Si XRD broader peak is dominant. The annealing time dependent XRD and micro–Raman TO peaks recognized that the nanocrystalline quality of Ge-O is gradually improved until certain annealing time afterwards it deteriorates again. The strain effect associated to the annealing time dependent crystalline quality that is expected to eventually improve the nano-crystalline properties. The annealed sample both Ge-O nanocrystal and Si (400) crystalline XRD peaks are dominant in which % of Si atom is decreased due to dominance of O atoms in the composition. The annealed SiGe coated microcavity both XRD and micro-Raman distinctive TO peaks for Ge-O and Si-Si bond are shown prominent. It is also characterized as annealing time dependent modulation of SiGe crystallinity. Either photo or electroluminescence can predominantly demonstrate the luminescence attributes of any crystalline size of semiconductors. To attributes SiGe nanostructure novel photonics, the dark current associated rectification ratio is most validated electrical characterization.

Reviewer #2: Presentation issues.

Section 2.1 Experimental Equipment

The authors do not give information on the type and brand (series) of the X-ray spectrometer used.

The same applies to the X-ray diffractometer.

The series of the Renishaw spectrometers is not presented either.

Globally.

The authors do not use Raman spectroscopy, but only photoluminescence one. RENISHAW Raman spectrometers usually enable studying both Raman scattering and photoluminescence using an appropriate channel.

In the text, abstract and figure captions, the terms such as Raman spectroscopy, Raman spectrum or Raman spectra, etc. must be replace with photoluminescence spectroscopy, photoluminescence spectrum (spectra), etc.

Pages 2 and 5. “Error! Reference source not found”

Figure 1. The inscription at the scale mark on the panel (a) is too small: it is difficult to see what is written.

Figures 2. The authors should indicate the mode of imaging on panels (a) and (b) or in the caption. Are the images secondary electron (SE) or backscattered electron (BSE) ones?

The authors should also reformulate the caption. Panel (c) does not show an image.

Besides, the table in the insert gives “quality percentage”. Maybe the authors mean percentage content. Is this atomic or weight percentage?

Further, the axes do not show titles and units of measurements. I can see that the energy of quantum in keV is given on the x-axis and the emission intensity is given on the y-axes (in some units, perhaps in pulse/sec/eV, as it is typical for AZtec (Oxford Instruments) EDX spectrometers).

The same applies to Figures 6 and 7.

Figures 3, 4 and 9. The units for the 2θ axes are not shown; I can see that they are degrees, yet the authors must indicate the units of measurements for all the axes.

In addition, there are no titles and units of measurements at the y-axes.

What diffractometer did the authors use? It should be indicated in Section 2.1 as well as the X-ray source (wavelength).

Finally, the authors did not indicate the temperature at which the luminescence was measured.

Text

Page 3, Section 2.2, paragraph 1. The surface cleaning process, including purity of cleaning agents, should be described in more details.

Obtaining of a clean Si surface is a key process for preparing a perfect structure.

Are the authors sure that the resultant Si surface was sufficiently clean after the described treatment and did not introduce uncontrolled impurities into the prepared structures?

What were the conditions in the chamber when etching cavities in Si?

“…spectrum of the sample is displayed in Figure 1(b), featuring a characteristic luminescence peak at 700 nm attributed to the silicon-based microcavity.” What are the rest peaks in the PL spectrum? As far as I can see, the sharp peak at ~547 nm is assigned to the TO phonon in crystalline Si (Raman shift ~ 520 1/cm), isn't it? The peak at 540 nm is 2TA in c-Si (Raman shift ~300 1/cm) and that at ~563 nm should be 2TO in c-Si (Raman shift ~1040 1/cm).

Are peaks related to SiO2 seen in the spectra? The Raman scattering bands associated with vibrations of the Si–O bonds in SiO2 might appear around 560 nm (Raman shift from ~800 to ~1200 1/cm) if silicon dioxide is present in the structure [DOI: 10.2138/am.2006.2075].

Page 4, paragraph 1. What was the gas composition of the atmosphere in the deposition chamber during the process? Was it clean enough in the chamber for the process to be conducted in correct conditions? Did the authors use some high-purity gases in the chamber? To what pressure was the chamber evacuated before the process?

Section 3.1, paragraph 1. “Figure 1(a) shows…” Figure 2(a)...

Page 5. Could the authors comment on the peaks at ~57° and ~75° in Figure 3(a) and the width of the band at 69°? If the former is GeO2, why did Ge oxidize when deposited? This should be briefly discussed in the text.

“The peaks at 31° and 69° correspond to the Si (200) and Si (400) crystal planes, indicating the presence of Ge and Si elements in the microcavity samples on the silicon substrate.”

Why Si peaks indicate the presence of Ge? Please, formulate the statement clearer.

Page 6. “Peaks at 31° and 69° correspond to the Si (200) and Si (400) crystal faces, indicating the presence of Ge and Si elements in the microcavity samples deposited on the silicon substrate, with possible incorporation of Ge elements.”

Again, why Si peaks indicate the presence of Ge? Please, formulate the statement clearer.

As far as I understand from Figure 4, initially, the authors see a Si(200) peak from the substrate, Si(400) from nanocrystals, two peaks from Ge and a reflection from GeO2(210) and some undiscussed reflection. After annealing for 10 min., they see the same pattern; in 20 min., the pattern is nearly the same, yet some new undiscussed reflections appear; in 30 min., the pattern mainly remains, yet Si(200) seems to have decreased, Ge (220) and (210) increased, Si (400) increased, some unidentified reflexes disappeared, whilst some intensified.

In 40 min., a strong narrow Si(400) reflex appears (poly-Si forms?). Ge(113) appears.

In 50 min., Si(400) (poly-Si?) go on growing. GeO2 decays.

In an hour, the reflexes weaken. (Why?)

Could the authors comment on this process by discussing what was happening to the Si, Ge and GeO2 during annealing.

Page 7. “…but with an additional sharp peak at this wavelength.”

This line is seen on panel (b).

“Thus, the sharp peak at 700 nm is due to localized luminescence from Ge-O bonds within these clusters.”

This statement does not seem entirely substantiated. The authors should give a justification for it.

What the authors mean when writing about the luminescence of the Ge–O bond? Do the authors mean the Raman band? GeO2 is transparent for the 532-nm laser.

Please, clarify this paragraph regarding the origin of the line peaking at 700 nm!

Figure 6. Perhaps, oxygen came on the surface due to film oxidation in the atmosphere during its moving to SEM. Otherwise, it is present in the SEM chamber. The treatment at 1000°C for 40 min should completely deoxidize the structure.

Figure 8. What is unannealing? Unannealed.

Section 3.3. “For the prepared germanium-coated silicon-based microcavity samples, PLD was used to deposit silicon onto a silicon target, followed by a layer of silicon structured over the germanium, creating a Ge-Si bilayer structure on the microcavity samples.”

The phrase is unclear. Please, reformulate it.

Section 3.4. The authors should explain where oxygen appears from in their experiment on annealing. Does the deposition chamber contain a lot of O2 gas in its atmosphere? Otherwise, the structure should deoxidize at such a high temperature. A few minutes should be enough to remove oxygen completely from the structure in vacuum or oxygen-free residual atmosphere.

My conclusion.

The manuscript may be considered for publication only after the authors have resolved all of the above issues.

6. PLOS authors have the option to publish the peer review history of their article (what does this mean? ). If published, this will include your full peer review and any attached files.

**Do you want your identity to be public for this peer review?** For information about this choice, including consent withdrawal, please see our Privacy Policy .

Reviewer #1: **Yes: ** Bablu Kumar Ghosh

Reviewer #2: No

---

## [Author Response · Author response to Decision Letter 1]

26 Dec 2024

We have made point-to-point revisions to the questions raised by the reviewers. We have also uploaded all the original data in the manuscript.

---

## [Decision Letter · Decision Letter 1]

10 Jan 2025

PONE-D-24-46249R1Luminescence Properties of Ge and Ge-Si Structures on Silicon-Based MicrocavitiesPLOS ONE

Dear Dr. Zhao,

Thank you for submitting your manuscript to PLOS ONE. After careful consideration, we feel that it has merit but does not fully meet PLOS ONE’s publication criteria as it currently stands. Therefore, we invite you to submit a revised version of the manuscript that addresses the points raised during the review process.

We look forward to receiving your revised manuscript.

Kind regards,

Mallikarjuna Reddy Kesama, Ph.D.

Academic Editor

PLOS ONE

Journal Requirements:

Additional Editor Comments:

The manuscript needs a thorough revision to address several issues related to typos, missing words, proper capitalization, and sentence structure for better clarity and readability.

Specific comments for figures and presentation:

General formatting: Ensure all graphs have the same font size and avoid using bold fonts in SEM images. Use a white background for all figures. Remove unnecessary boxes around titles (e.g., "unannealed" or "annealing at different timings").

Figure-specific Comments:

Figure 2c: Missing information for the x-axis (Energy in KeV) and y-axis (Intensity(a.u.)).

Figure 3a & 3b: Y-axis information is missing for Figure 3a. the font style differs between Figures 3a and 3b when compared to Figure 2. Maintain uniformity.

Figure 4: Clearly label the y-axis (e.g., Intensity (a.u.)). The inset in figure 4e is unclear and needs to be replotted for better visibility.

Figure 5a: The line thickness is inconsistent compared to other lines in the figure. Ensure uniformity.

Figures 6a, 6b, and 6c: Replace bold fonts with normal fonts and use a white background.

Figure 7: Use normal font for labels (a, b, c). X-axis and y-axis information is missing.

Figure 8a: Y-axis information is missing.

Figure 9: Y-axis information is missing in all parts.

Figure 11: Revise this figure to match the updated standards.

Axis Intervals: Across Figures 1 to 11, the y-axis has too many interval points. Use larger intervals to make the data presentation clearer.

Reviewer comments:

Address all reviewer comments thoroughly.

Overall Presentation: The current presentation of data gives the impression of inexperience in publishing articles. Revise figures and text to improve the quality and professionalism of the manuscript.

Kindly address these suggestions and ensure all corrections are made in both text and figures.

Reviewers' comments:

Reviewer's Responses to Questions

**Comments to the Author**

1. If the authors have adequately addressed your comments raised in a previous round of review and you feel that this manuscript is now acceptable for publication, you may indicate that here to bypass the “Comments to the Author” section, enter your conflict of interest statement in the “Confidential to Editor” section, and submit your "Accept" recommendation.

Reviewer #1: All comments have been addressed

Reviewer #2: (No Response)

2. Is the manuscript technically sound, and do the data support the conclusions?

Reviewer #1: Yes

Reviewer #2: Yes

3. Has the statistical analysis been performed appropriately and rigorously?

Reviewer #1: N/A

Reviewer #2: N/A

4. Have the authors made all data underlying the findings in their manuscript fully available?

Reviewer #1: Yes

Reviewer #2: Yes

5. Is the manuscript presented in an intelligible fashion and written in standard English?

Reviewer #1: Yes

Reviewer #2: Yes

6. Review Comments to the Author

Reviewer #1: The manuscript wording has been revised. Anyway, the Raman spectroscopy is not photoluminescence. Please address the real methodology and marked it properly. The high spectral resolution of a Raman spectrometer can be useful for PL spectroscopy, especially when the emission spectra have narrow bands or lines. Though it is not mentioned properly.

Reviewer #2: 1) What do the authors mean when writing “Characterization images…”? Maybe, the authors could just use “characterization” in the captions?

2) The word “photoluminescence” should not be capitalized unless it begins a sentence.

Please check your manuscript carefully for typos!

3)

170 “In this experiment,

171 silicon wafers were soaked in a solution of 95% concentration of

172 ethanol and acetone mixed 1:1 for five minutes to remove surface

173 organic matter, then submerged in deionized water and cleaned in an

174 ultrasonic cleaner for 15 min.”

Please, indicate the purity of the components.

4)

166 “Photoluminescence (PL) spectra were measured at

167 room”

… room temperature.

5)

205 “The deposited samples were then placed in a

206 tube furnace where they were annealed at 1000°C for varying

207 durations to study the luminescent properties of the samples with

208 different deposited elements and annealing times.”

The pressure and the atmosphere composition should be presented in the text. The pressure in the deposition chamber before the process is also absent in the text.

6)

234 “Peaks at 27° and 45° correspond to the Ge (110) and

235 Ge (220) crystal planes, respectively, and are sharp and narrow,

236 indicating high crystallinity. The peaks at 31° and 69° correspond to

237 the Si (200) and Si (400) crystal planes. Indicating the presence of

238 Ge and Si elements in the microcavity samples on the silicon

239 substrate.”

From my point of view, the statement in the text has become even less comprehensible.

The sentence written in the response to my comment is much clearer: “The XRD pattern reveals the coexistence of Ge (110) and Ge (220) crystal planes, as well as Si (200) and Si (400) crystal planes on the surface of the silicon-based microcavity. This indicates the presence of both Ge and Si elements in the microcavity samples deposited on silicon substrates.”

Sorry, I cannot understand the remark 此部分仅是回复。Google translates it as “This is a very important part of our work.”

7)

239 The 57° and 75° germanium dioxide peaks may result

240 from low peak intensities due to a small amount of oxidation on the

241 surface of the sample after preparation or during testing.

The phrase is poorly formulated. The phrase “a small amount of oxidation” is unclear.

8)

248 based clusters at the edge of the microcavityError! Reference source not found..

From the authors’ response.

9) “Could the authors comment on this process by discussing what was happening to the Si, Ge and

GeO2 during annealing.

My response As the annealing time varies, the gradual improvement in the quality of Ge-O nanocrystals

can be observed through the TO peak of XRD and Raman spectroscopy, until a certain annealing time is

reached where the Ge-O cluster structure is disrupted. The strain effect associated with annealing time is

expected to ultimately enhance the properties of the nanocrystals. The XRD peaks of both germanium

oxide nanocrystals and silicon (400) crystals dominate in the annealed samples, with the percentage of

silicon atoms decreased due to the dominance of oxygen atoms in the composition.此部分仅是回复。”

I recommend including this discussion in the manuscript.

10) “Please, clarify this paragraph regarding the origin of the line peaking at 700 nm!

My response�A group has employed a quantum dot confinement model to calculate the energy gap

structure of germanium oxide nanocrystal clusters, and utilized the Monte Carlo method to simulate the

PL spectrum and the corresponding size distribution of the germanium oxide nanocrystal clusters. The

calculated results are in agreement with our experimental findings, leading us to analyze that this is due

to the specific luminescent peaks exhibited by small-sized Ge-O clusters formed through the combination

of small-sized Ge and O.此部分仅是回复。”

In my opinion, this should be written in the text. Otherwise, the statements are quite speculative.

In addition, it is advisable to present the simulation results as a supporting material.

11) “15. Figure 6. Perhaps, oxygen came on the surface due to film oxidation in the atmosphere during

its moving to SEM. Otherwise, it is present in the SEM chamber. The treatment at 1000°C for 40

min should completely deoxidize the structure.

My response The sample surface is populated with many small-sized germanium crystal clusters. When

the sample is processed under annealing conditions, the germanium crystals on the surface react with

oxygen atoms. After the oxygen atoms diffuse into the germanium layer, they coalesce into Ge-O clusters

above the oxide layer. However, annealing for a duration longer than a certain threshold can disrupt the

cluster structure of Ge-O. 此部分仅是回复。”

Unfortunately, the description of the annealing conditions is absent in the text. They should be presented.

12) “18. Section 3.4. The authors should explain where oxygen appears from in their experiment on

annealing. Does the deposition chamber contain a lot of O2 gas in its atmosphere? Otherwise, the

structure should deoxidize at such a high temperature. A few minutes should be enough to remove

oxygen completely from the structure in vacuum or oxygen-free residual atmosphere.

My response The deposition process is carried out under vacuum conditions, while the annealing process

is conducted in an atmospheric environment. The source of oxygen during annealing is the oxygen

present in the atmosphere, which reacts with the surface Ge under high temperature to form a cluster

structure of Ge-O.此部分是回复.”

This should be stated in the text.

7. PLOS authors have the option to publish the peer review history of their article (what does this mean? ). If published, this will include your full peer review and any attached files.

**Do you want your identity to be public for this peer review?** For information about this choice, including consent withdrawal, please see our Privacy Policy .

Reviewer #1: No

Reviewer #2: No

---

## [Author Response · Author response to Decision Letter 2]

7 Feb 2025

My Response to Reviewers

PLOS ONE, PONE-D-24-46249R1

Luminescence Properties of Ge and Ge-Si Structures on Silicon-Based Microcavities

1. Additional Editor Comments:

The manuscript needs a thorough revision to address several issues related to typos, missing words, proper capitalization, and sentence structure for better clarity and readability.

Specific comments for figures and presentation:

General formatting: Ensure all graphs have the same font size and avoid using bold fonts in SEM images. Use a white background for all figures. Remove unnecessary boxes around titles (e.g., "unannealed" or "annealing at different timings").

My response�I have made the corresponding modifications, please refer to lines 232、361、389、and 470 for details.

Figure-specific Comments:

Figure 2c: Missing information for the x-axis (Energy in KeV) and y-axis (Intensity(a.u.)).

My response�I have made the corresponding modifications, please refer to lines 233 for details.

Figure 3a & 3b: Y-axis information is missing for Figure 3a. the font style differs between Figures 3a and 3b when compared to Figure 2. Maintain uniformity.

My response�I have made the corresponding modifications, please refer to lines 255 for details.

Figure 4: Clearly label the y-axis (e.g., Intensity (a.u.)). The inset in figure 4e is unclear and needs to be replotted for better visibility.

My response�I have made the corresponding modifications, please refer to lines 282、283 and 284 for details.

Figure 5a: The line thickness is inconsistent compared to other lines in the figure. Ensure uniformity.

My response�I have made the corresponding modifications, please refer to lines 343 for details.

Figures 6a, 6b, and 6c: Replace bold fonts with normal fonts and use a white background.

My response�I have made the corresponding modifications, please refer to lines 361 for details.

Figure 7: Use normal font for labels (a, b, c). X-axis and y-axis information is missing.

My response�I have made the corresponding modifications, please refer to lines 389 and 390 for details.

Figure 8a: Y-axis information is missing.

My response�I have made the corresponding modifications, please refer to lines 394 for details.

Figure 9: Y-axis information is missing in all parts.

My response�I have made the corresponding modifications, please refer to lines 423、424 and 425 for details.

Figure 11: Revise this figure to match the updated standards.

My response�I have made the corresponding modifications, please refer to lines 470 for details.

Axis Intervals: Across Figures 1 to 11, the y-axis has too many interval points. Use larger intervals to make the data presentation clearer.

My response�I have made the corresponding modifications, please refer to lines 233 255 282 283 284 343 344 345 361 390 394 423 424 425 452 453 454 and 470 for details.

2. Reviewer Questions:

Reviewer #1: The manuscript wording has been revised. Anyway, the Raman spectroscopy is not photoluminescence. Please address the real methodology and marked it properly. The high spectral resolution of a Raman spectrometer can be useful for PL spectroscopy, especially when the emission spectra have narrow bands or lines. Though it is not mentioned properly.

My response The text has been revised——In this study, a Renishaw Raman spectrometer was used to test the PL spectroscopy of the sample, which had an excitation light of 532 nm, a spectral detection range of 547-900 nm, and an excitation power of up to 50 mW� “Please refer to lines 167 to 170 of the paper for details”.

Reviewer #2: 1) What do the authors mean when writing “Characterization images…”? Maybe, the authors could just use “characterization” in the captions?

My response The characterization images have been modified to 'characterization' in the text.“Please refer to lines 216、256、362、391、395、471 of the paper for details”.

2) The word “photoluminescence” should not be capitalized unless it begins a sentence.

Please check your manuscript carefully for typos!

My response Change some photoluminescence that is not at the beginning to lowercase letters.“Please refer to lines 185、217、247、257、289、294、310、385、396、429、431、436、478 and 543 of the paper for details”.

3)

170 “In this experiment,

171 silicon wafers were soaked in a solution of 95% concentration of

172 ethanol and acetone mixed 1:1 for five minutes to remove surface

173 organic matter, then submerged in deionized water and cleaned in an

174 ultrasonic cleaner for 15 min.”

Please, indicate the purity of the components.

My response The text has been revised—In this experiment, silicon wafers were immersed in a mixed solution of 30 milliliters of 95% concentrated ethanol and 30 milliliters of 98% concentrated acetone for 5 minutes to remove surface organics. Afterward, the silicon wafers were cleaned with deionized water using an ultrasonic cleaner for 15 minutes�“Please refer to lines 173 to 178 of the paper for details”.

4)

166 “Photoluminescence (PL) spectra were measured at

167 room”

… room temperature.

My response Room temperature has been added to the main text. Please refer to lines 51of the paper for details.

5)

205 “The deposited samples were then placed in a

206 tube furnace where they were annealed at 1000°C for varying

207 durations to study the luminescent properties of the samples with

208 different deposited elements and annealing times.”

The pressure and the atmosphere composition should be presented in the text. The pressure in the deposition chamber before the process is also absent in the text.

My response Add annealing conditions. Please refer to lines 209 to 214 of the paper for details.

6)

234 “Peaks at 27° and 45° correspond to the Ge (110) and

235 Ge (220) crystal planes, respectively, and are sharp and narrow,

236 indicating high crystallinity. The peaks at 31° and 69° correspond to

237 the Si (200) and Si (400) crystal planes. Indicating the presence of

238 Ge and Si elements in the microcavity samples on the silicon

239 substrate.”

From my point of view, the statement in the text has become even less comprehensible.

The sentence written in the response to my comment is much clearer: “The XRD pattern reveals the coexistence of Ge (110) and Ge (220) crystal planes, as well as Si (200) and Si (400) crystal planes on the surface of the silicon-based microcavity. This indicates the presence of both Ge and Si elements in the microcavity samples deposited on silicon substrates.”

Sorry, I cannot understand the remark. Google translates it as “This is a very important part of our work.”

My response The text has been revised—The XRD pattern reveals the coexistence of Ge (110) and Ge (220) crystal planes, as well as Si (200) and Si (400) crystal planes on the surface of the silicon-based microcavity� Please refer to lines 240 to 242 of the paper for details.

7)

239 The 57° and 75° germanium dioxide peaks may result

240 from low peak intensities due to a small amount of oxidation on the

241 surface of the sample after preparation or during testing.

The phrase is poorly formulated. The phrase “a small amount of oxidation” is unclear.

My response The text has been revised—In the XRD pattern, there are also two low peaks located at 57° and 75° , indicating the generation of weak germanium dioxide crystal phase during laser etching� Please refer to lines 243 to 245 of the paper for details.

8)

248 based clusters at the edge of the microcavity Error! Reference source not found..

From the authors’ response.

My response Add reference information. Please refer to lines 252 of the paper for details.

9) “Could the authors comment on this process by discussing what was happening to the Si, Ge and GeO2 during annealing.

My response As the annealing time varies, the gradual improvement in the quality of Ge-O nanocrystals can be observed through the TO peak of XRD and Raman spectroscopy, until a certain annealing time is reached where the Ge-O cluster structure is disrupted. The strain effect associated with annealing time is expected to ultimately enhance the properties of the nanocrystals. The XRD peaks of both germanium oxide nanocrystals and silicon (400) crystals dominate in the annealed samples, with the percentage of silicon atoms decreased due to the dominance of oxygen atoms in the composition.

I recommend including this discussion in the manuscript.

My response It has been added in the body—As the annealing time changes, it can be observed through XRD and Raman spectroscopy that the TO peak of Ge-O gradually increases, and the crystallization performance improves until a certain annealing time is reached. The Ge-O cluster structure is destroyed, and the crystallization performance deteriorates. In the annealed samples, XRD peaks of germanium oxide nanocrystals and silicon (400) crystals dominate, and the percentage of silicon atoms decreases due to the dominant position of oxygen atoms in the composition; Please refer to lines 318 to 326 of the paper for details.

10) “Please, clarify this paragraph regarding the origin of the line peaking at 700 nm!

My response�A group has employed a quantum dot confinement model to calculate the energy gap structure of germanium oxide nanocrystal clusters, and utilized the Monte Carlo method to simulate the PL spectrum and the corresponding size distribution of the germanium oxide nanocrystal clusters. The calculated results are in agreement with our experimental findings, leading us to analyze that this is due to the specific luminescent peaks exhibited by small-sized Ge-O clusters formed through the combination of small-sized Ge and O.”

In my opinion, this should be written in the text. Otherwise, the statements are quite speculative.

In addition, it is advisable to present the simulation results as a supporting material.

My response It has been added in the body—Huang W Q had employed a quantum dot confinement model to calculate the energy gap structure of germanium oxide nanocrystal clusters, and utilized the Monte Carlo method to simulate the PL spectrum and the corresponding size distribution of the germanium oxide nanocrystal clusters. The calculated results are in agreement with our experimental findings; Please refer to lines 302 to 307 of the paper for details; Reference added on line 579, line 581.

11) “15. Figure 6. Perhaps, oxygen came on the surface due to film oxidation in the atmosphere during its moving to SEM. Otherwise, it is present in the SEM chamber. The treatment at 1000°C for 40 min should completely deoxidize the structure.

My response The sample surface is populated with many small-sized germanium crystal clusters. When the sample is processed under annealing conditions, the germanium crystals on the surface react with oxygen atoms. After the oxygen atoms diffuse into the germanium layer, they coalesce into Ge-O clusters above the oxide layer. However, annealing for a duration longer than a certain threshold can disrupt the cluster structure of Ge-O.

Unfortunately, the description of the annealing conditions is absent in the text. They should be presented.

My response It has been added in the body—The sample surface is populated with many small-sized germanium crystal clusters. When the sample is processed under annealing conditions, the germanium crystals on the surface react with oxygen atoms. After the oxygen atoms diffuse into the germanium layer, they coalesce into Ge-O clusters above the oxide layer. However, annealing for a duration longer than a certain threshold can disrupt the cluster structure of Ge-O Please refer to lines 331 to 337 of the paper for details.

12) “18. Section 3.4. The authors should explain where oxygen appears from in their experiment on annealing. Does the deposition chamber contain a lot of O2 gas in its atmosphere? Otherwise, the structure should deoxidize at such a high temperature. A few minutes should be enough to remove oxygen completely from the structure in vacuum or oxygen-free residual atmosphere.

My response The deposition process is carried out under vacuum conditions, while the annealing process is conducted in an atmospheric environment. The source of oxygen during annealing is the oxygen present in the atmosphere, which reacts with the surface Ge under high temperature to form a cluster structure of Ge-O.

This should be stated in the text.

My response It has been added in the body—The source of oxygen during annealing is the oxygen present in the atmosphere, which reacts with the surface Ge under high temperature to form a cluster structure of Ge-O Please refer to lines 444 to 447 of the paper for details.

---

## [Editor Report · Decision Letter 2]

11 Feb 2025

PONE-D-24-46249R2Luminescence Properties of Ge and Ge-Si Structures on Silicon-Based MicrocavitiesPLOS ONE

Dear Dr. Zhao,

Thank you for submitting your manuscript to PLOS ONE. After careful consideration, we feel that it has merit but does not fully meet PLOS ONE’s publication criteria as it currently stands. Therefore, we invite you to submit a revised version of the manuscript that addresses the points raised during the review process.

We look forward to receiving your revised manuscript.

Kind regards,

Mallikarjuna Reddy Kesama, Ph.D.

Academic Editor

PLOS ONE

Journal Requirements:

Additional Editor Comments: I would like to provide you with an additional opportunity to address the minor revisions and update the manuscript as per the reviewers' comments.

---

## [Author Response · Author response to Decision Letter 3]

14 Mar 2025

Thank you for your reply. We've made changes to the article by removing the text about funding from the manuscript and adding the relevant title information for the supporting information file. The funder, Prof. Shihua Zhao, played a role in the research design, data analysis, publication decision-making, and manuscript revision in this paper.

---

## [Decision Letter · Decision Letter 3]

25 Mar 2025

Luminescence Properties of Ge and Ge-Si Structures on Silicon-Based Microcavities

PONE-D-24-46249R3

Dear Dr. Shihua Zhao,

We’re pleased to inform you that your manuscript has been judged scientifically suitable for publication and will be formally accepted for publication once it meets all outstanding technical requirements.

Kind regards,

Mallikarjuna Reddy Kesama, Ph.D.

Academic Editor

PLOS ONE

Reviewers' comments:

Reviewer's Responses to Questions

**Comments to the Author**

1. If the authors have adequately addressed your comments raised in a previous round of review and you feel that this manuscript is now acceptable for publication, you may indicate that here to bypass the “Comments to the Author” section, enter your conflict of interest statement in the “Confidential to Editor” section, and submit your "Accept" recommendation.

Reviewer #1: All comments have been addressed

Reviewer #2: All comments have been addressed

2. Is the manuscript technically sound, and do the data support the conclusions?

Reviewer #1: Yes

Reviewer #2: (No Response)

3. Has the statistical analysis been performed appropriately and rigorously?

Reviewer #1: Yes

Reviewer #2: (No Response)

4. Have the authors made all data underlying the findings in their manuscript fully available?

Reviewer #1: Yes

Reviewer #2: (No Response)

5. Is the manuscript presented in an intelligible fashion and written in standard English?

Reviewer #1: No

Reviewer #2: (No Response)

6. Review Comments to the Author

Reviewer #1: The authors addressed all issues relating to publishing the manuscript. Now the paper is ready for publishing.

Reviewer #2: The authors took into account the reviewer’s comments and revised the manuscript accordingly.

I advise publishing the revised manuscript.

7. PLOS authors have the option to publish the peer review history of their article (what does this mean? ). If published, this will include your full peer review and any attached files.

**Do you want your identity to be public for this peer review?** For information about this choice, including consent withdrawal, please see our Privacy Policy .

Reviewer #1: No

Reviewer #2: No

---

## [Editor Report · Acceptance letter]

PONE-D-24-46249R3

PLOS ONE

Dear Dr. Zhao,

I'm pleased to inform you that your manuscript has been deemed suitable for publication in PLOS ONE. Congratulations! Your manuscript is now being handed over to our production team.

Kind regards,

on behalf of

Dr. Mallikarjuna Reddy Kesama

Academic Editor

PLOS ONE